# Lysis of arterial thrombi by perfusion of N,N'-Diacetyl-L-cystine (DiNAC)

**Dongjune Kim**[1], **Susan M. Shea**[2], **David N. Ku**[1] *

**1** G.W. Woodruff School of Mechanical Engineering, Georgia Institute of Technology, Atlanta, Georgia, United States of America, **2** School of Medicine, Washington University in St. Louis, St. Louis, Missouri, United States of America

* david.ku@me.gatech.edu

**Data Availability Statement:** All relevant data are within the paper and its Supporting information files.

**Funding:** This study was funded by L.P. Huang Chair.

## Abstract

The search persists for a safe and effective agent to lyse arterial thrombi in the event of acute heart attacks or strokes due to thrombotic occlusion. The culpable thrombi are composed either primarily of platelets and von Willebrand Factor (VWF), or polymerized fibrin, depending on the mechanism of formation. Current thrombolytics were designed to target red fibrin-rich clots, but may be not be efficacious on white VWF-platelet-rich arterial thrombi. We have developed an *in vitro* system to study the efficacy of known and proposed thrombolytic agents on white clots formed from whole blood in a stenosis with arterial conditions. The agents and adjuncts tested were tPA, ADAMTS-13, abciximab, N-acetyl cysteine, and N,N'-Diacetyl-L-cystine (DiNAC). Most of the agents, including tPA, had little thrombolytic effect on the white clots. In contrast, perfusion of DiNAC lysed thrombi as quickly as 1.5 min, which ranged up to 30 min at lower concentrations, and resulted in an average reduction in surface area of 71 ± 20%. The clot burden was significantly reduced compared to both tPA and a saline control ($p<0.0001$). We also tested the efficacy of all agents on red fibrinous clots formed in stagnant conditions. DiNAC did not lyse red clots, whereas tPA significantly lysed red clot over 48 h ($p<0.01$). These results lead to a novel use for DiNAC as a possible thrombolytic agent against acute arterial occlusions that could mitigate the risk of hyper-fibrinolytic bleeding.

## Introduction

An acute ischemic stroke is a catastrophic event resulting from the occlusion of an artery supplying blood to the brain. Approximately 700,000 cases of ischemic stroke occur in the United States yearly, resulting in a financial burden of more than US$70 billion [1, 2]. Additionally, stroke survivors and their families are devastated by the disabling aftereffects, and many elderly patients admit to fearing survival more than death [1, 3]. Therefore, treatment of an acute stroke in a safe, efficacious, and expedient manner is a current major priority for the field.

The goal of ischemic stroke treatment is the expeditious clearance of the occluding thrombus to regain perfusion of the downstream vessel bed (reperfusion). The current clinical

**Competing interests:** The authors have declared that no competing interests exist.

standard for stroke treatment is the use of the intravenous (IV) tissue plasminogen activator (tPA) [1, 4]. tPA was first discovered in 1947 [5], and was tested with different types of *in vitro* assays (e.g. a fibrin plate [6], a circulating plasma system [7]). The high thrombolytic efficacy of tPA found in *in vitro* assays led to an *in vivo* study [8], a small clinical study [9], and large clinical trials [10–12]. Still, tPA is the only Food and Drug Administration (FDA) approved thrombolytic agent in the United States [13]. However, patients must receive treatment within 3–4.5 h of the onset of stroke symptoms, and many individuals have contraindications, such as a recent surgery or bleeding [1, 4]. A clinical study found that tPA has a limited recanalization rate of less than 30% [14]. Another clinical trial demonstrated a 30% relative risk reduction versus the placebo in stroke patients, but ultimately, there was no statistically significant improvement in the overall mortality with use of tPA [13].

tPA is a serine protease that catalyzes the conversion of plasminogen to its active form, plasmin, in the vicinity of a hemostatic plug. Plasmin then cleaves fibrin, thus breaking down or lysing the thrombus [4, 15]. However, thrombi causing an arterial occlusion may not be fibrin-rich, and thus tPA may not be effective in this setting. Arterial thrombi form under very high shear stress hemodynamics prior to occlusion, and are structurally very different from a fibrin gel [16]. The two major contributors to occlusive stroke are ischemia from thrombotic occlusion of an atherosclerotic carotid stenosis (either *in situ* or thromboembolic) or the formation of embolic clots within the heart, such as in patients with atrial fibrillation (cardioembolic) [4]. Because of the shear-dependent mechanisms of thrombus formation, these situations likely produce thrombi of very different compositions: high shear *white* von Willebrand Factor (VWF)-platelet thrombi in the case of a thromboembolic stroke and low shear *red* fibrin clots in the case of a cardioembolic stroke [4, 17–20]. The composition of the clot may determine the efficacy of a thrombolytic drug. As arterial thrombi are formed under high shear conditions and are VWF-platelet rich, we hypothesize that tPA may not be the most efficacious treatment for ischemic strokes of a thromboembolic origin. In addition, tPA has a high rate of bleeding complications due to the induction of a hyperfibrinolytic state, which deter its clinical use [21–23]. Therefore, a great need remains for a safe and efficacious thrombolytic drug for the treatment of ischemic stroke and myocardial infarction.

We tested the efficacy of potential novel thrombolytic agents and tPA on VWF-platelet-rich white thrombi. Additional potential lytic agents were selected based on mechanistic efficacy in the literature. ADAMTS-13 is the protease that cleaves VWF, the major protein responsible for capture of platelets under high shear rates, and therefore potentiation of white thrombosis [20, 24, 25]. Denorme et al. showed the potential use of ADAMTS-13 in acute ischemic stroke in a FeCl-induced injury mice model [26]. Abciximab inhibits platelet thrombus formation by blocking the glycoprotein IIb/IIIa through an antibody. Thus, it is considered as an antithrombotic drug in general, but also has been used as an adjuvant thrombolytic agent [27, 28]. N-acetylcysteine (NAC) has been shown to inhibit platelet thrombus formation via the degradation of plasma VWF multimers [29]. Furthermore, De Lizarrondo et al. demonstrated that NAC can be used in the thrombolysis of FeCl-induced thrombi [30]. Hastings and Ku (ISTH 2017) reported that NAC lyses platelet-rich thrombi better than do the above-mentioned thrombolytic agents [31]. However, we have since found a high variability between NAC batches and the presence of N,N'-Diacetyl-L-cystine (DiNAC) in high-efficacy batches. Thus, we also explored the use of DiNAC as a thrombolytic agent. DiNAC is the disulfide dimer of NAC that has been studied for its anti-atherosclerotic effects [32, 33] but not for its use in thrombolytic therapy. We find that DiNAC has the potential to be a highly efficient, novel thrombolytic agent.

## Material and methods

### Collagen coating

The stenotic glass test sections (inner diameter = 1.5 mm) used in this study were made by the Chemistry and Biochemistry Glass Shop at Georgia Institute of Technology. The % stenosis by diameter reduction ranged from 60% to 80%. Fibrillar equine collagen (type I; Chrono-Log Corporation, Havertown, PA) was diluted 9:1 in NaCl (Sigma-Aldrich, St. Louis, MO) and incubated in the test section at the stenosis for 24 h in a warm, moist environment [20, 34].

### Blood collection

Porcine blood was collected at a local abattoir (Holifield farm, Covington, GA) and did not require the Animal Care and Use Committees (IACUC) approval. Immediately following slaughter, the blood was poured into 3.5 U/mL heparin (Thermo Fisher Scientific, Waltham, MA) or 3.2% sodium citrate (Sigma-Aldrich, St. Louis, MO). Whole blood was gently agitated using an Orbit LS shaker (Laboratory Supply Network, Atkinson, NH) until perfusion and used in experiments as soon as possible the same day as collection.

### Syringe perfusion

Lightly heparinized whole blood was perfused through the stenotic test section using a syringe pump. The flow rate was set so that the initial wall shear rate in the stenosis was 3,500 s$^{-1}$. A pressure transducer was connected in-line upstream of the stenosis (Fig 1A). A dissecting

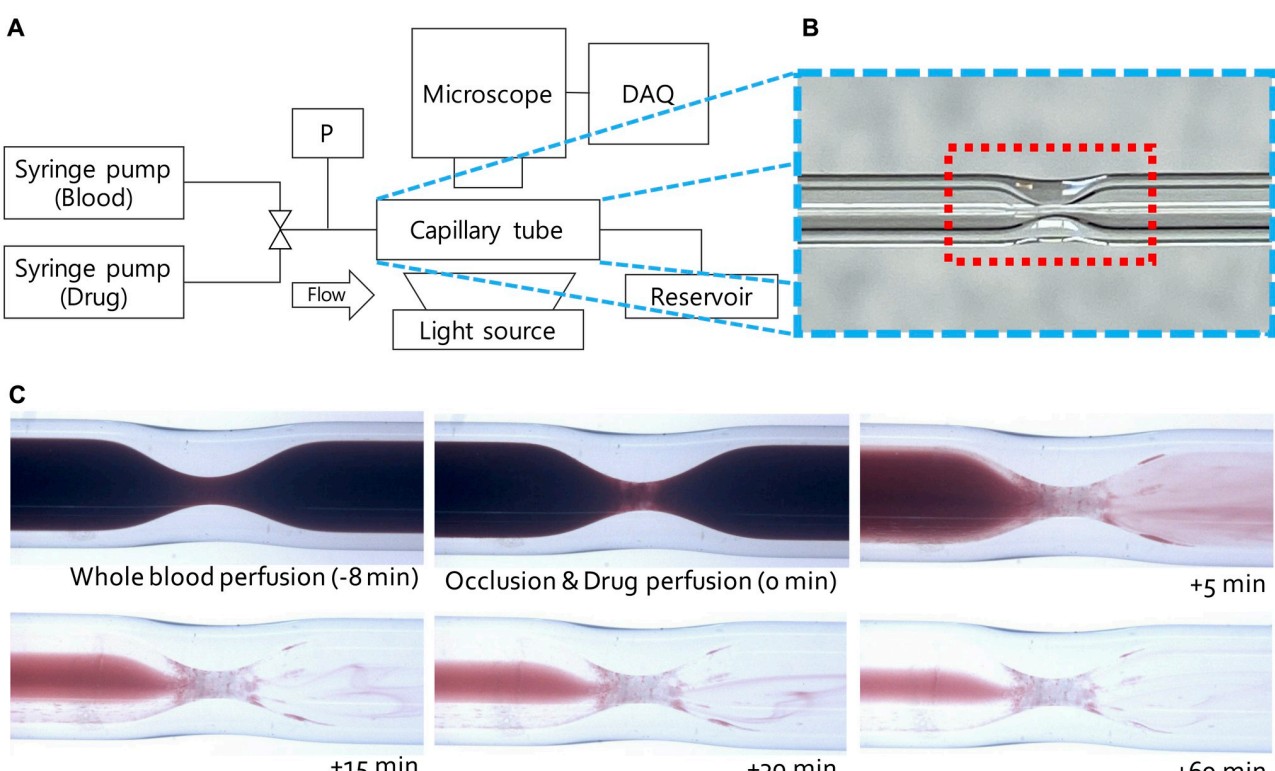

**Fig 1. The *in vitro* perfusion system for creating platelet-rich, occlusive thrombi under arterial high shear rates, followed by perfusion of known and potential lytic agents. A** Schematic of the arterial flow setup. **B** Close-up of the glass capillary tube with stenosis, which is coated with fibrillar collagen prior to perfusion. The red box denotes the region of interest. **C** Thrombus formation and subsequent perfusion with a phosphate-buffered saline (PBS) control, showing no lysis at the end of the experiment.

microscope with a camera (PCO-Tech Incorporated, Romulus, MI) was used to capture images in real-time during perfusion. Blood perfusion continued until the upstream pressure increased by 30 mmHg (equivalent to an arterial pressure head *in vivo*) as a result of platelet thrombus formation in the stenotic section. Perfusion with a syringe pump (constant flow rate over constant pressure head) ensured that the thrombi were never fully occlusive, allowing subsequent perfusion for the induction of lysis.

## Pharmacologic agents

Recombinant human tPA (Sigma-Aldrich, St. Louis, MO), recombinant human ADAMTS-13 (MyBioSource, San Diego, CA), abciximab (ReoPro was kindly provided by Dr. Kevin Maher at Emory/CHOA), NAC (Thermo Fisher Scientific, Waltham, MA), and DiNAC (Cayman Chemical, Ann Arbor, MI) or control (PBS) treatments were perfused for an hour at a flow rate of 1 ml/min. Agent solutions were made by dissolution or dilution in PBS. The *in vitro* flow system setup and image acquisition are shown in Fig 1, and the concentrations of each agent are detailed in Table 1. Real-time image capture (5 frames per second) of the stenosis continued throughout treatment perfusion.

## Red clot formation and lysis

Platelet rich plasma (PRP) was made by separating citrated whole blood via gravity over a 2 h period and collecting the supernatant. Separation via gravity was employed instead of centrifugation to avoid platelet damage and activation. Citrated whole blood or PRP was then recalcified with $CaCl_2$ to a final $[Ca^{2+}]$ of 10 mM [35], and 200 ul was transferred into 500 ul centrifuge tubes and allowed to clot and retract for 30 min. The clot was then incubated with 100 ul of either agent or control solution. The treatment solution was exchanged by removing the top 100 ul and replacement with fresh solution at 3, 6, 12, 24, and 48 h of incubation. The weight of the tube was measured immediately post-clot formation and at each timepoint after removal of old solution and prior to the replacement with fresh solution.

## Computational fluid dynamics analysis

Computational fluid dynamics (CFD) was used to calculate the drag force acting on the thrombi formed in the stenosis under flow. Simulations were performed using Ansys 19.1 (Ansys Inc, PA, USA). Whole blood was assumed to be Newtonian fluid of 3.5 cP and flow was presumed as laminar, incompressible, steady, continuous, and isothermal due to the low Reynolds number (Re = 16). The capillary tube was modeled with no-slip walls and 1 ml/min flow rate was applied at the inlet with zero-pressure at the outlet, reflecting experimental conditions. Mesh convergence was achieved at 3.8 million tetrahedral cells yielding a residual error of $10^{-9}$.

**Table 1. Agent concentration and replicate number.**

| Agent | Concentration | Number of replicates |
|---|---|---|
| DiNAC | 0.02 mM, 0.2 mM, 2 mM, 20 mM, 20[#] mM | 8, 8, 9, 8, 8 |
| NAC | 2 mM, 20 mM | 7, 5 |
| tPA | 20 $\mu$g/mL | 4 |
| ADAMTS-13 | 1 $\mu$g/mL | 4 |
| Abciximab | 35 $\mu$g/mL | 4 |
| PBS | - | 10 |

## Data analysis

The thrombi surface area (colored green) was calculated using manual pixel counting in the open-source GNU Image Manipulation Program (GIMP, Version 2.10.8, 1995–2018). Surface area reduction was calculated by % pixel reduction versus the occlusion image. One-way analysis of variance (ANOVA) with the Tukey's post hoc test was used to test for statistical differences between groups using Prism 9 (GraphPad Software, San Diego, CA). Statistical significance was set at $p < 0.05$. Data are displayed as mean with error bars denoting standard error of the mean (SEM).

## Results

### DiNAC is a more efficacious thrombolytic agent than NAC against arterial white clots

Initial perfusion with NAC showed promising lytic results [31]. However, replicate perfusion experiments with solutions made with NAC yielded inconsistent results with minimal lysis. To further investigate this contradiction, we analyzed the initial solutions with high thrombolytic efficacy using mass spectrometry and found a large peak of DiNAC (S1 and S2 Figs). We then hypothesized that DiNAC, and not NAC, could be the potential thrombolytic agent capable of platelet-rich white thrombi lysis, which would also account for the inconsistencies in reports of efficacy of NAC in the field [36, 37]. To test this hypothesis, we modified our stenotic capillary tube model, which potentiates VWF- and platelet-rich white thrombi in an arterial setting on a collagen-coated stenosis [16, 20, 34] to test lysis of occlusive white thrombi (Fig 1) and perfused low (2 mM) and high (20 mM) concentrations of NAC and DiNAC solutions. Assuming a Poiseuille flow and uniform thrombi growth from the glass wall, > 80% occlusive white thrombi by area will result in a 30 mmHg pressure gradient. DiNAC, but not NAC, achieved thrombolysis over 60 minutes of perfusion (Fig 2A–2D). 2 mM and 20 mM DiNAC lysed most of the thrombi (> 95%) within 14 min and 1.5 min, respectively (Fig 2A and 2B). In contrast, NAC perfusion resulted in minimal lysis (Fig 2C and 2D, S1 Movie). DiNAC showed a significantly higher thrombus surface area reduction than NAC after 60 min of perfusion (Fig 2E and 2F) for both concentrations (2 mM $p < 0.001$, 20 mM, $p < 0.01$).

A video of the DiNAC lysis is provided as "S2 Movie." Large pieces of the thrombus break off dramatically as it lyses, but the embolizing sections are sometimes tethered by long string-like structures that eventually release.

### DiNAC dose response

To confirm the efficacy of DiNAC, we tested the dose response at increasing concentrations of 0.02, 0.2, 2, and 20 mM. The thrombus area reduction was dependent on the concentration of DiNAC (Fig 3). The reduction was the highest at 2 mM (71 ± 20%) and slightly decreased at the highest concentration of 20 mM (59 ± 12%), though this was not significantly different ($p = 0.86$). Perfusion with the lowest concentration of 0.02 mM DiNAC was not significantly different from the control, but 0.2 mM showed a significantly decreased thrombus area (46 ± 15%, $p = 0.02 < 0.05$). We also found that the 20 mM DiNAC solution was acidic (pH ≈ 2), while 2 mM DiNAC was neutral (pH = 7). Considering the practical use of the DiNAC as a thrombolytic in a clinical setting, we neutralized the 20 mM DiNAC with sodium bicarbonate to a final pH = 7. Neutralization of 20 mM DiNAC reduced the variance from 47% to 12% in surface area reduction (44 ± 47% vs 59 ± 12%, $p = 0.77$, Fig 3).

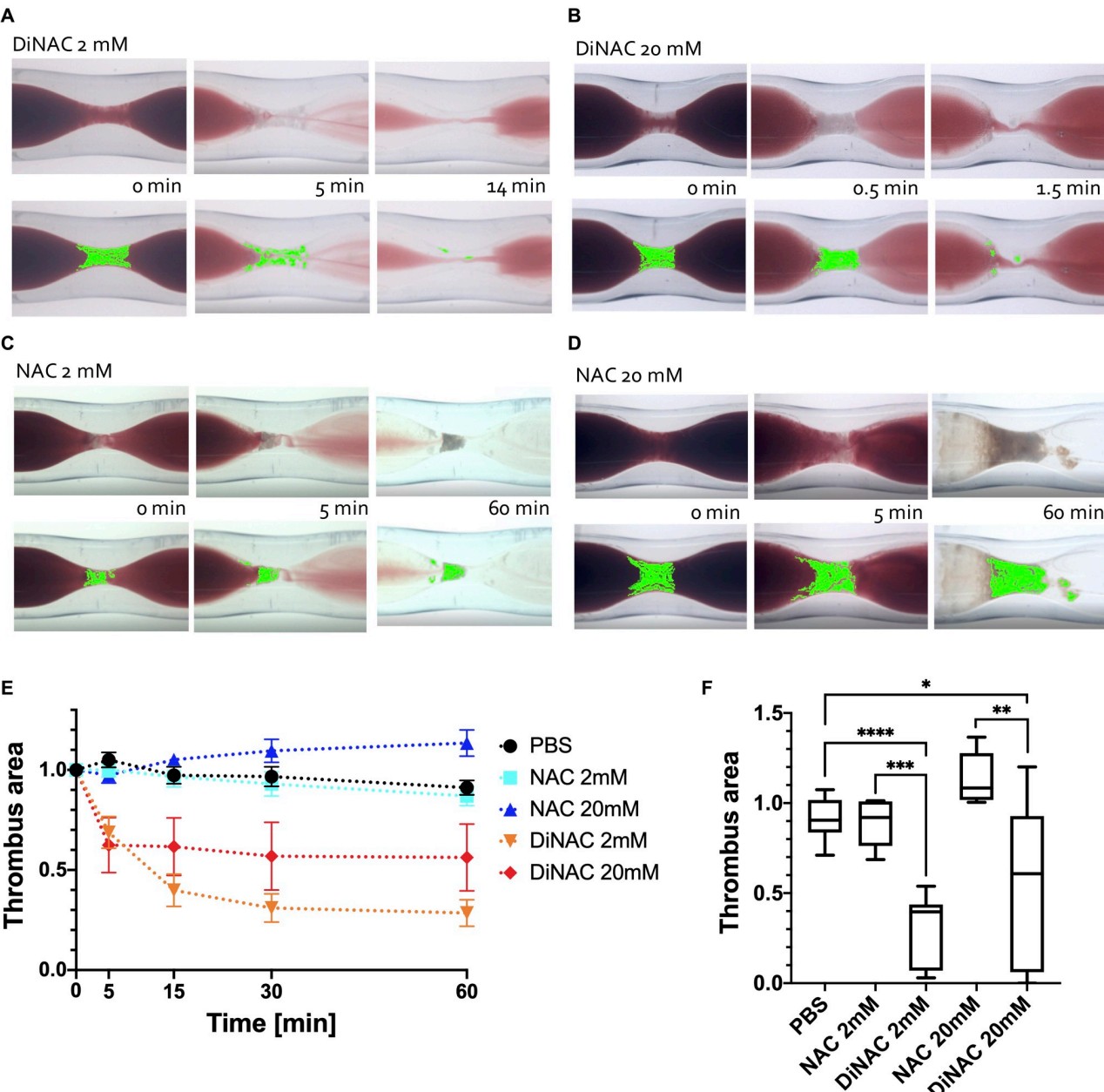

**Fig 2. Perfusion with DiNAC and NAC.** Thrombus surface area was determined by pixel counting and is shown in paired images below originals, with thrombus area highlighted in green. **A** 2 mM DiNAC perfusion showing complete (> 95% surface area reduction) lysis in 14 min. **B** 20 mM DiNAC perfusion showing complete lysis in 1.5 min. **C** 2 mM NAC perfusion, with minimal lysis (< 20% surface area reduction) after 60 min. **D** 20 mM NAC perfusion after 60 min, again with minimal lysis even at increased concentration. **E** Thrombus area reduction over time. Phosphate-buffered saline (PBS) is included as a negative control (black line). **F** Thrombus area after 60 min perfusion with the indicated agent (x-axis). 2 mM and 20 mM DiNAC cause significantly more lysis than the control, while neither concentration of NAC was more efficacious than PBS. DiNAC was also significantly different than NAC at each concentration. * $p < 0.05$; ** $p < 0.01$; *** $p < 0.001$; **** $p < 0.0001$.

## The effect of other thrombolytic agents on platelet-rich white clot

Perfusion with phosphate-buffered saline (PBS; negative control) for 60 min reduced the thrombus surface area by 9 ± 12% (Fig 1C and see S3 Movie). Perfusion for 60 minutes with 0.02 mg/ml tPA reduced the thrombus surface area by 23±4% (Fig 4A and see S4 Movie),

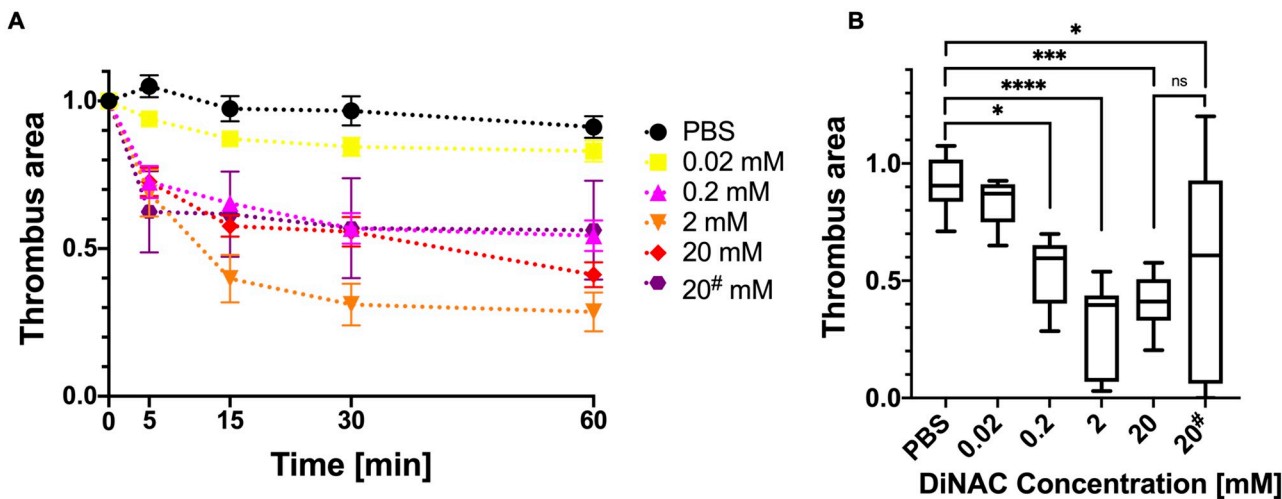

**Fig 3. DiNAC dosage response with concentrations of 0.02, 0.2, 2, and 20 mM.** The # denotes the acidic DiNAC 20 mM solution. **A** Thrombus area reduction over time. The control is shown in black. 0.02 mM DiNAC was not different from the control (yellow), and 2 mM DiNAC (orange) had the greatest efficacy **B** Thrombus area after 60 min perfusion with increasing concentrations of DiNAC. Concentrations of 0.2 mM and greater were significantly different from the control. Neutralization of DiNAC mitigated variability and increased the surface area reduction (20 vs. 20* mM). * $p < 0.05$; *** $p < 0.001$; **** $p < 0.0001$.

demonstrating a weak thrombolytic effect relative to DiNAC. Perfusion with ADAMTS-13 reduced the thrombus area by 19±8% (Fig 4B and see S5 Movie), and perfusion with Abciximab resulted in a reduction in thrombus area of 22±11% (Fig 4C and see S6 Movie). None of these three agents were significantly different than the control (Fig 4E and 4D: tPA, $p = 0.43$; ADAMTS-13, $p = 0.52$; Abciximab, $p = 0.31$).

## DiNAC does not lyse fibrin clots

Clinical use of tPA is associated with bleeding risks due to the induction of a systemic hyper-fibrinolytic state [21–23]. We generated fibrinous red clots by re-calcifying citrated whole blood or PRP under stagnant conditions and applied agents to quantify thrombolytic efficacy. PRP was included to determine if any platelet-specific interactions by agents may be occurring as the increase in platelet concentration creates clots with greater platelet density. Only tPA demonstrated significant thrombolytic efficacy on clots formed in these conditions (Fig 5). tPA significantly reduced both clot volume and weight over the course of 48 h ($p < 0.01$) compared to the control. In contrast, the other agents (DiNAC, NAC, and ADAMTS-13) did not cause any reduction in clot size nor volume. Abciximab was expected to have a less bleeding risk unless used with anticoagulation therapy [38] and excluded due to a limited supply. Coagulated red clots generally form under low shear rate conditions such as in veins or bleeding. The lack of lytic efficacy on clots formed under stagnant conditions suggests that DiNAC may mitigate the life-threatening risk of hemorrhage associated with current tPA thrombolytic therapy.

## Platelet-rich thrombi elongate and break during DiNAC thrombolysis

Treatment with DiNAC lyses the intact white thrombus by causing it to break apart in fragments. The fragments were typically towards the center of the lumen, and away from the wall, and often remained tethered to the main body of the thrombus. The string-like tails stretched,

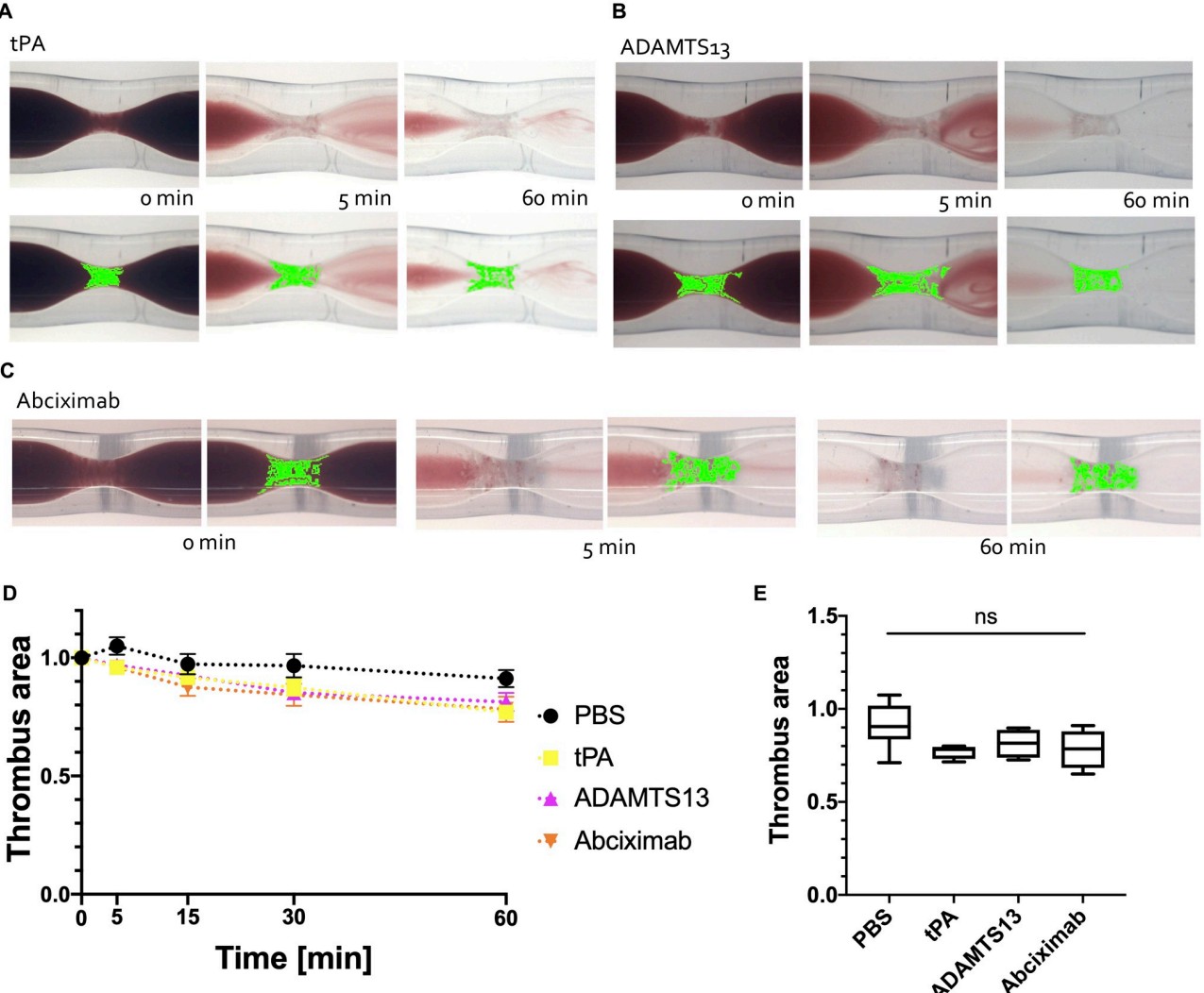

**Fig 4. Thrombolysis with the other agents.** Thrombus area is highlighted in green. **A** tPA, **B** ADAMTS-13, and **C** abciximab perfusion showed minimal lysis after 60 min. **D** Thrombus area reduction over time. **E** Thrombus area after 60 min perfusion with the indicated agent (x-axis). Perfusion of tPA, ADAMTS-13, and abciximab had no effect on white clot, with no differences from the control.

yet in some cases persisted for several minutes, before eventual breakage, releasing the fragments to wash downstream (Fig 6A and see S7 Movie).

The flow conditions during the thrombi lysis via fragmentation were modeled using CFD (Fig 6B) to quantify the shear stresses and drag forces on the fragments. High-velocity jet-like flow was seen in the stenosis with recirculation downstream (Fig 6C). The elongated thrombi and fragment tails are visualized in gray in Fig 6C. Shear rate was maximal at the stenosis reaching over 15,000 $s^{-1}$ while shear rates of approximately 4,000–8,000 $s^{-1}$ acted on the surface of thrombus downstream (Fig 6D). The total drag force was 380 nN for a simulated thinner fragment and 780 nN for a simulated larger fragment (Fig 6E). Thus, the net attachment force of the tethers needs to be greater than 380,000 pN, suggesting that the thrombus is held by many thousands of bonds.

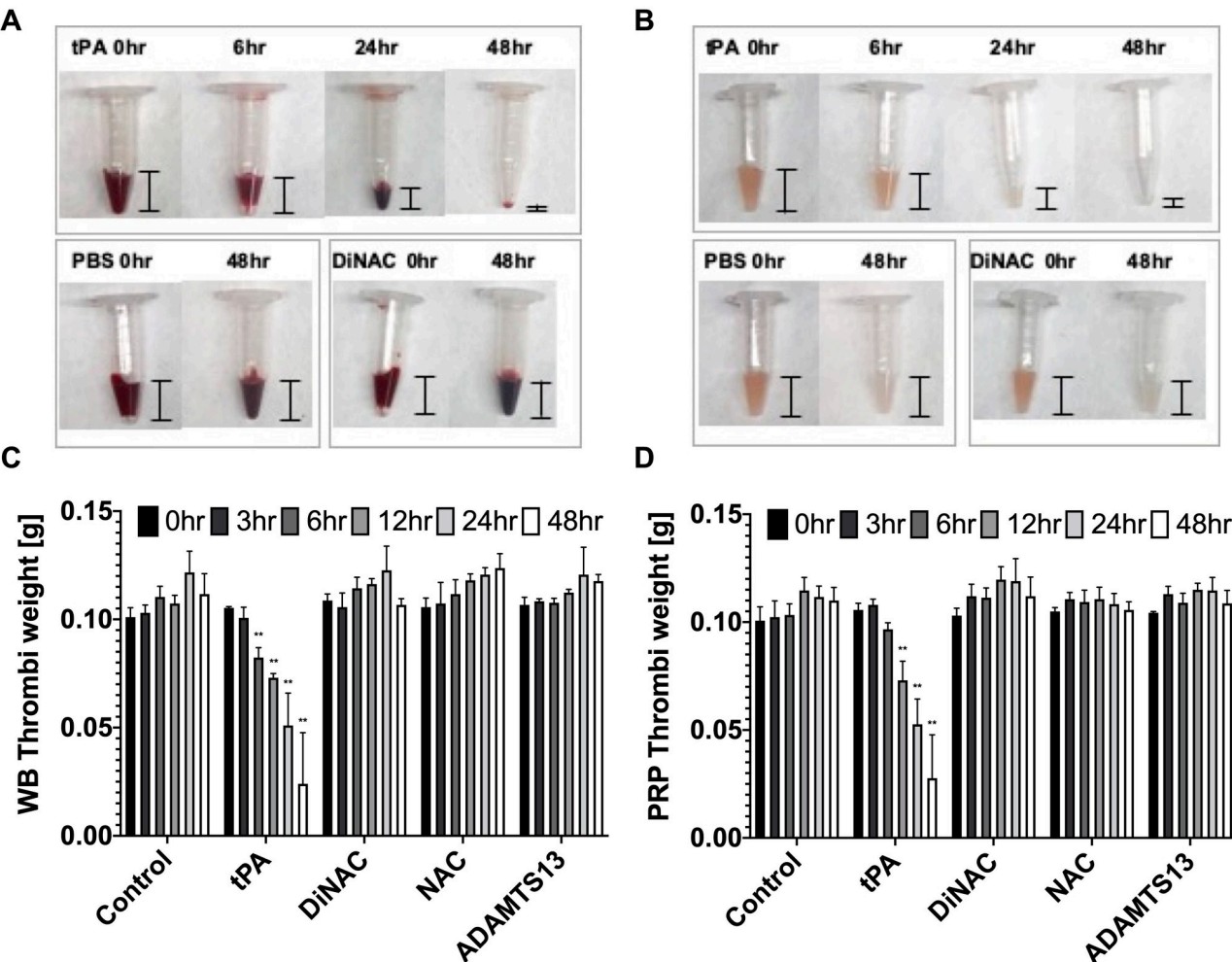

**Fig 5. Lysis results on fibrin clots formed under stagnant conditions (n = 3 per agent).** DiNAC and NAC were tested at a concentration of 2 mM. **A** Whole blood clot lysis over 48 h. tPA showed a large decrease in clot volume. **B** PRP clot lysis over 48 h. tPA again showed a large decrease in clot volume. **C** and **D** Whole blood and PRP clot weight change over 48 h. tPA lysed clots resulting in a weight change significantly different from baseline after 6 h for whole blood and 12 h for PRP (26% and 36% reduction, respectively, $^{**}p < 0.01$). DiNAC, NAC, and ADAMTS-13 had no effect on red clots, with no differences from baseline nor control.

## Discussion

DiNAC, but not NAC, demonstrated the ability to completely lyse platelet-rich thrombi under perfusion in an arterial setting. These thrombi formed in the setting of arterial shear rates over fibrillar collagen are composed of VWF and platelets [16, 19]. Other agents, such as tPA, ADAMTS-13, and abciximab showed a limited ability to achieve VWF-platelet thrombus lysis and did not reduce thrombus surface area any no more than the PBS control.

tPA achieves thrombolysis by converting plasminogen to plasmin, which in turn, cleaves fibrin. This process requires the presence of endogenous plasminogen and takes some time in the local setting for the kinetics of biochemical reactions to occur [39, 40]. We were therefore concerned about the efficacy of tPA against VWF-platelet white clots where fibrin may be less structural to the clot, and in the arterial setting where high flow rates may inhibit enzymatic reactions due to the dominance of convection. Indeed, tPA has produced mixed results against ischemic strokes from clinical arterial occlusions [41]. Post-analysis of occlusive thrombi from

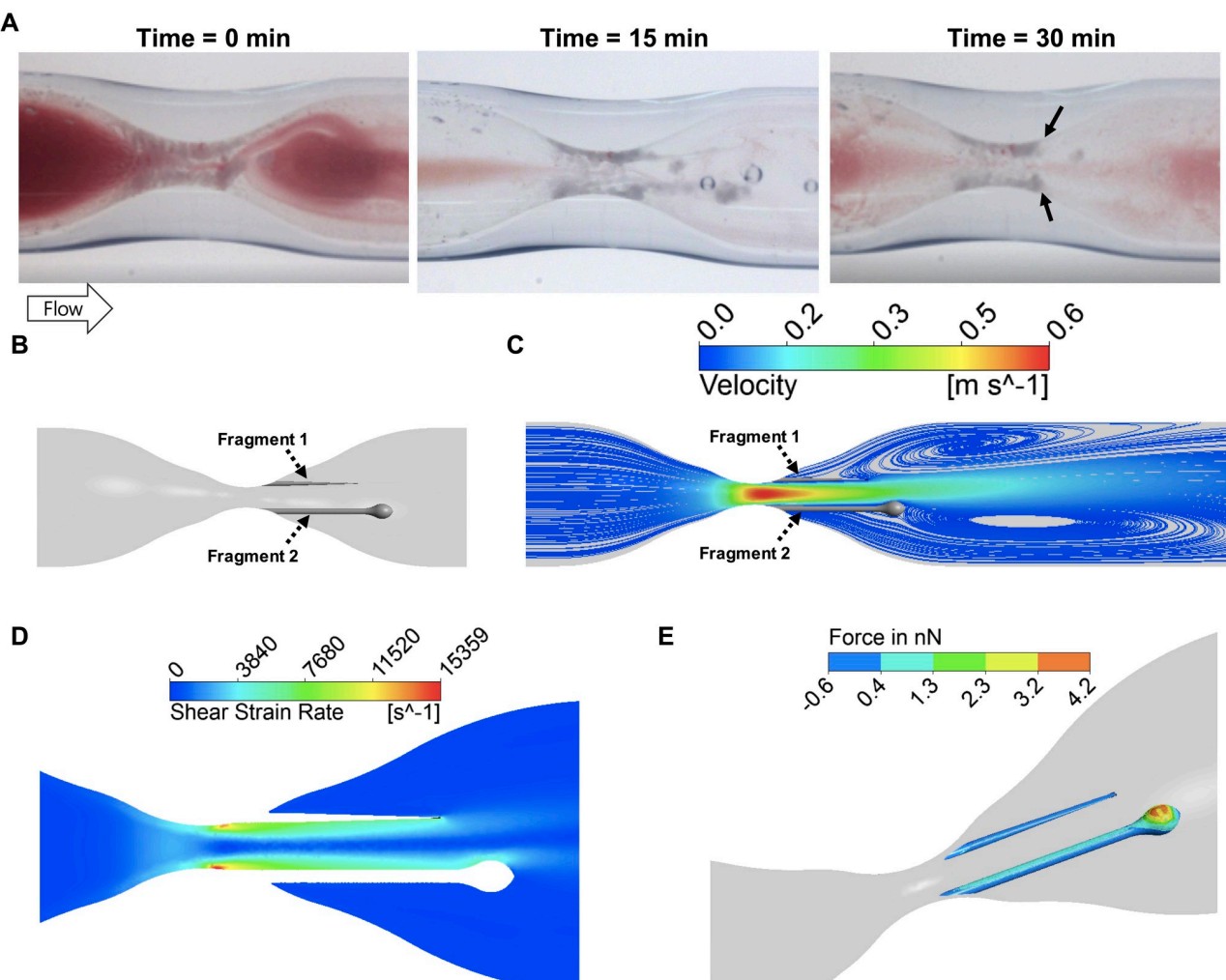

**Fig 6. Simulation of flow and force through the stenosis using Computational Fluid Dynamics (CFD). A** Structure of a thrombus during elongation and breakage at 0, 15, and 30 minutes of DiNAC perfusion. Black arrows denote points of tether breakage. **B** Computer model of the attached thrombus fragments at 15 min. **C** Velocity streamlines showing a jet-like flow and recirculation downstream of the elongated thrombus colored in gray. **D** Shear strain rate around the thrombus. A maximum shear of 15,000 s⁻¹ (red) was observed in the throat of stenosis. **E** Drag force acting on the thrombi surface. The thrombus 2 fragment experiences a maximum force of > 4 nN.

patients have been unable to determine a correlation between origin and composition. Marder et al. examined 25 thrombi retrieved from the cerebral circulation of patients and did not find consistency among the suspected etiologies and compositions, but they demonstrated a large variety of structural components [4, 42]. However, because of flow cessation, coagulation cascades that may not have caused the occlusive event are likely triggered post-occlusion in the vicinity of the original culprit thrombus.

We did observe some lysis of VWF- and platelet-rich thrombi under tPA perfusion. tPA may lyse these thrombi more efficaciously with the addition of circulating plasminogen to the system. Tersteeg et al. suspected that plasmin is a possible back-up enzyme for ADAMTS-13. They found that plasmin indeed possesses some ability to degrade platelet-VWF complexes [43]. We were surprised that ADAMTS-13 did not seem to have a thrombolytic effect on the occlusive thrombi. ADAMTS-13 is a metalloprotease that reduces VWF adhesion by cleavage of ultra-large VWF multimers [21, 44] and porcine VWF has same cleavage site (A2) as

human VWF [45]. ADAMTS-13 has been shown to have some antithrombotic activity *in vivo* with potential as a thrombolytic agent [21, 26]. ADAMTS-13 may require the VWF molecule be under tension to work. In the static clot, the normal length VWF may not be forcefully stretched for ADAMTS-13 cleavage [46, 47]. Steric hindrance might also prevent penetration of ADAMTS-13 (molecular weight 190 kDa) into the thrombus compared with a small molecule like DiNAC (molecular weight 324 Da). In a study by Crescente et al., both tPA and ADAMTS-13 were shown to reduce thrombus size (by 53.2% and 62.3%, respectively) after a 60 min treatment [21]. That model used the $FeCl_3$ injury to induce the thrombus *in vivo* and also involved treatment with a higher concentration of ADAMTS-13 (4 $\mu$g/mL), which may account for the differences in our results. Our trials were limited in concentration by the level of ADAMTS-13 in normal plasma (1 $\mu$g/mL) [48] due to the prohibitively high cost of ADAMTS-13. The cost could ultimately limit its clinical use. Abciximab has been shown to have a positive effect as an adjuvant to thrombolytic therapy, as it inhibits the heightened platelet activation and aggregation observed in patients treated with tPA [27, 28]. We did not see a significant thrombolytic effect with abciximab alone. In the future, testing abciximab together with other agents may be informative. Lysis/reperfusion studies post myocardial infarction have been previously performed in domestic pigs, suggesting this model is relevant [49]. However, species differences between human and porcine blood response may exist.

NAC is known to reduce mucin multimers, and VWF is strikingly similar to mucins in structure [29, 50]. NAC is currently used as a treatment for chronic obstructive lung disease and acetaminophen overdose. Chen et al. demonstrated VWF degradation by NAC, leading to our use of this agent in the present study [29]. De Lizarrondo et al. showed the potential use of NAC as a thrombolytic agent using an *in vivo* mice model [30], but they found limited reperfusion with NAC treatment alone ($< 40\%$).

We found high thrombolytic efficacy of DiNAC. The marked ability of DiNAC, and not NAC, to lyse platelet thrombi is a surprising discovery. DiNAC is a disulfide dimer of two NAC monomers, that has previously been studied for its anti-atherosclerotic effects [32, 33]. DiNAC showed a dose-response effect for concentrations of 0.02, 0.2, and 2 mM, but showed slightly decreased thrombolytic efficacy at 20 mM. The 20 mM DiNAC solution was very acidic and had high variability in efficacy. The variability was attenuated by neutralization. Thrombolysis by DiNAC created macroscopic fissures in the thrombus body, followed by the formation of tethered fragments and finally an eventual break with tolerable micro emboli passing downstream. The initial fracture is away from the wall, suggesting that the point of lysis is not a collagen bond. The strings are presumably VWF since the length is on the order of a millimeter. We quantified the drag force on thrombi fragments to be between 380 and 780 nN. To hold the thrombi in place would require approximately 3,800 to 7,800 bonds from either GpIba [51] or GpIIb/IIIa at ~100 pN/bond, consistent with our prior estimates of the number of bonds need to grow the thrombus [52]. The mechanism of thrombolysis of white clot by DiNAC is an interesting topic as DiNAC does not contain a free thiol, which is essential for NAC to cleave VWF multimers [29] or any other blood components that has sulfide bonds which could limit delivery of NAC to VWF. Instead, DiNAC has a sulfide bond that can be reversed to two thiols [53]. Thus, DiNAC may be delivered into the clot and converted to two NACs before reacting with VWF, or there could be a new reaction that induces switching between two disulfide bonds. We leave this topic for a future study.

Clots may form via different mechanisms, thus with different resulting morphologies, and different relative content of fibrin v. VWF. Thus, different types of clot may require different thrombolytic agents. Hyper-fibrinolytic states induced by thrombolytic treatment are associated with increased mortality in many disease etiologies. We therefore performed studies of *in vitro* red clot lysis under static conditions over 48 h with the same agents. Only tPA showed

significant thrombolytic efficacy on fibrin clots, dissolving the clot with an approximate front speed of 3.5 μm/min that is comparable to the other *in vitro* fibrinolytic assays [54, 55]. The other agents, including DiNAC, did not show any significant red clot lysis. Thus, DiNAC may not cause severe bleeding, which has limited the use of tPA in patients due to iatrogenic hyper-fibrinolysis. The limited effect of DiNAC on PRP clots, which would have greater platelet concentration than those formed with WB, suggests that DiNAC is not reacting with platelets, but may be interacting with VWF or the VWF-platelet bond to achieve thrombolysis.

This study is limited in some ways. Heparin was used to block coagulation in whole blood samples used for creation of white clots to minimize the cross-effects against VWF and platelets seen frequently with citrate. Heparin is an indirect thrombin inhibitor, and therefore allows high shear platelet-rich thrombi to form uninhibited but may have other small downstream effects on clot formation. Porcine whole blood was used and may have species differences with human whole blood, though we expect these to be minimal [56]. ADAMTS-13 and abciximab may have had a limited thrombolytic efficacy in porcine blood compared to human blood. The thrombi were viewed top-down through the test section, and the 2-dimensional area was estimated by pixel counts that does not measure the volume distribution. Other 3-D high-resolution imaging techniques in real time may reveal the location of fractures to provide different insights into thrombus lytic mechanisms. Our experiments were not performed on mature clots that may have contracted, reducing porosity. Small reductions in surface area over 30 minutes were observed in the negative controls, which may at least partially represent concomitant contraction during agent perfusion. Although there was mixing between whole blood and agents right after switching the syringe, the lack of added plasma may have hindered the potential effect of tPA (plasminogen) or ADAMTS-13 (calcium ions). This model also requires prohibitively expensive amounts of ADAMTS-13. In the future, an *in vivo* model could be used to impose more physiologic conditions and examine toxicity. A mouse *in vivo* and microfluidic *in vitro* models that require less volume could be used to further study ADAMTS-13 lysis. In this study, we focused on reporting the efficacy of agents, while future work exploring mechanisms behind the thrombolytic behavior of DiNAC may be a community endeavor. Interactions between VWF-DiNAC and VWF-platelet-DiNAC are left for a future study. We also leave the challenge of delivery to completely occluded sections for future investigation.

## Conclusions

DiNAC, and not NAC, was highly efficacious in the lysis of VWF-platelet-rich thrombi created in a stenotic coronary artery analog system perfused under high shear stress. Other agents demonstrated a limited ability to lyse arterial thrombi in this setting, including tPA and ADAMTS-13. DiNAC was unable to lyse red fibrinous clot in a stagnant setting, while tPA was highly efficacious in the latter system. These results indicate the possibility of DiNAC as an effective thrombolytic agent against arterial occlusions, with the potential to mitigate life-threatening side effects of hemorrhage associated with current thrombolytic therapies.

## Supporting information

**S1 Fig. Mass spectrometry result of DiNAC.**
(TIF)

**S2 Fig. Mass spectrometry result of NAC.**
(TIF)

**S1 Movie. High shear white thrombus formation and subsequent NAC perfusion.**
(MP4)

**S2 Movie. High shear white thrombus formation and subsequent lysis by DiNAC perfusion.**
(MP4)

**S3 Movie. High shear white thrombus formation and subsequent PBS perfusion.**
(MP4)

**S4 Movie. High shear white thrombus formation and subsequent tPA perfusion.**
(MP4)

**S5 Movie. High shear white thrombus formation and subsequent ADAMTS13 perfusion.**
(MP4)

**S6 Movie. High shear white thrombus formation and subsequent abciximab perfusion.**
(MP4)

**S7 Movie. Thrombi elongating, breaking, and fragmenting during perfusion of DiNAC.**
(MP4)

## Acknowledgments

D.K., S.M.S., and D.N.K designed the research and drafted the manuscript. D.K. and S.M.S. performed the experiments. D.K and S.M.S. did data analysis. All authors discussed the results and commented on the manuscript.

## Author Contributions

**Conceptualization:** Dongjune Kim, Susan M. Shea, David N. Ku.

**Data curation:** Dongjune Kim.

**Funding acquisition:** David N. Ku.

**Investigation:** Dongjune Kim, Susan M. Shea.

**Methodology:** Dongjune Kim, Susan M. Shea.

**Project administration:** David N. Ku.

**Supervision:** David N. Ku.

**Validation:** Dongjune Kim.

**Visualization:** Dongjune Kim.

**Writing – original draft:** Dongjune Kim.

**Writing – review & editing:** Dongjune Kim, Susan M. Shea, David N. Ku.

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
