## [Decision Letter · Decision Letter 0]

22 Dec 2020

PONE-D-20-37780

Lysis of arterial thrombi by perfusion of N,N’-Diacetyl-L-cystine (DiNAC)

PLOS ONE

Dear Dr. Ku,

Thank you for submitting your manuscript to PLOS ONE. After careful consideration, we feel that it has merit but does not fully meet PLOS ONE’s publication criteria as it currently stands. Therefore, we invite you to submit a revised version of the manuscript that addresses the points raised during the review process.

We look forward to receiving your revised manuscript.

Kind regards,

Christoph E Hagemeyer, PhD

Academic Editor

PLOS ONE

Journal Requirements:

2. Please note that PLOS does not permit references to “data not shown.” Authors should provide the relevant data within the manuscript, the Supporting Information files, or in a public repository. If the data are not a core part of the research study being presented, we ask that authors remove any references to these data.

3. To comply with PLOS ONE submission guidelines, in your Methods section, please provide additional information regarding your statistical analyses. For more information on PLOS ONE's expectations for statistical reporting, please see https://journals.plos.org/plosone/s/submission-guidelines.#loc-statistical-reporting.

Reviewers' comments:

Reviewer's Responses to Questions

**Comments to the Author**

1. Is the manuscript technically sound, and do the data support the conclusions?

Reviewer #1: Partly

Reviewer #2: Partly

2. Has the statistical analysis been performed appropriately and rigorously? 

Reviewer #1: I Don't Know

Reviewer #2: Yes

3. Have the authors made all data underlying the findings in their manuscript fully available?

Reviewer #1: Yes

Reviewer #2: No

4. Is the manuscript presented in an intelligible fashion and written in standard English?

Reviewer #1: Yes

Reviewer #2: Yes

5. Review Comments to the Author

Reviewer #1: Please refer to attached reviewer comments for a more comprehensive review of the manuscript. Major points for consideration include; needing to expand upon the performed statistical analysis and performing the experiments with appropriately matched pharmacological agents to blood species.

Reviewer #2: The study is overall very interesting. The efficacy of thrombolysis changing according to blood clot composition is an important question for which clear answer is difficult to get. In vitro study comparing different clot type and different thrombolytic such as this study is therefore very important for the field. However, there are some important lack of data to support some claims, in particular regarding the difference of efficacy between diNAC and NAC. I recommend major revision and would be grandly appreciated if the authors could answer several points :

1) The results section start with

« Initial perfusion with NAC showed promising lytic results. However, replicate perfusion experiments with solutions made with NAC purchased from a variety of vendors yielded nconsistent results with minimal lysis. To further investigate this contradiction, we analyzed the initial solutions with high thrombolytic efficacy using mass spectrometry and found a large peak of DiNAC (data not shown) »

The author should provide the details of these experiments, if several batch of NAC have been tested, the details of the batches should be provided, along with their respective efficacy and the analysis that were performed, including mass spectrometry analysis.

Alternatively, the authors should remove this paragraph from the results section and present in a first figure a mass spectrometry analysis of the NAC (Thermo Fisher Scientific, Waltham, MA) and the DiNAC (Cayman Chemical, Ann Arbor, MI).

Alternative methods showing the difference between NAC and DiNAC are also acceptables.

2) The supplemental videos are much appreciated. However, one video from each group should be provided.

3) Additional experiments showing a difference of activity from NAC and DiNAC would be appreciated. If DiNAC has a reduction potential higher than NAC, it could be measured in vitro with basic molecular biology methods. It could be tested on soluble vWF maybe as well ? Analysed with SDS-PAGE with non denaturing condition ?

Alternatively, the authors should state what they have tested and explain why no difference could be seen between NAC and DiNAC.

4) The discussion should provide hypothesis on why DiNAC is more potent than NAC in vitro.

Minor comments :

There several spelling mistakes that should be corrected.

6. PLOS authors have the option to publish the peer review history of their article (what does this mean?). If published, this will include your full peer review and any attached files.

Reviewer #1: No

Reviewer #2: No

---

## [Author Response · Author response to Decision Letter 0]

22 Jan 2021

Please see Response to Reviewers letter at the end of this PDF

---

## [Decision Letter · Decision Letter 1]

9 Feb 2021

Lysis of arterial thrombi by perfusion of N,N’-Diacetyl-L-cystine (DiNAC)

PONE-D-20-37780R1

Dear Dr. Ku,

We’re pleased to inform you that your manuscript has been judged scientifically suitable for publication and will be formally accepted for publication once it meets all outstanding technical requirements.

Kind regards,

Christoph E Hagemeyer, PhD

Academic Editor

PLOS ONE

Additional Editor Comments (optional):

Reviewers' comments:

Reviewer's Responses to Questions

**Comments to the Author**

1. If the authors have adequately addressed your comments raised in a previous round of review and you feel that this manuscript is now acceptable for publication, you may indicate that here to bypass the “Comments to the Author” section, enter your conflict of interest statement in the “Confidential to Editor” section, and submit your "Accept" recommendation.

Reviewer #1: All comments have been addressed

Reviewer #2: All comments have been addressed

2. Is the manuscript technically sound, and do the data support the conclusions?

Reviewer #1: Yes

Reviewer #2: Yes

3. Has the statistical analysis been performed appropriately and rigorously? 

Reviewer #1: Yes

Reviewer #2: Yes

4. Have the authors made all data underlying the findings in their manuscript fully available?

Reviewer #1: Yes

Reviewer #2: Yes

5. Is the manuscript presented in an intelligible fashion and written in standard English?

Reviewer #1: Yes

Reviewer #2: Yes

6. Review Comments to the Author

Reviewer #1: All comments have been reflected in the manuscript accordingly. Recommended manuscript to be accepted for publication.

Reviewer #2: The authors have taken into consideration the different comments and provided the additinal data requested.

7. PLOS authors have the option to publish the peer review history of their article (what does this mean?). If published, this will include your full peer review and any attached files.

Reviewer #1: No

Reviewer #2: No

---

## [Editor Report · Acceptance letter]

16 Feb 2021

PONE-D-20-37780R1 

Lysis of arterial thrombi by perfusion of N,N’-Diacetyl-L-cystine (DiNAC) 

Dear Dr. Ku:

I'm pleased to inform you that your manuscript has been deemed suitable for publication in PLOS ONE. Congratulations! Your manuscript is now with our production department. 

Kind regards, 

on behalf of

Dr. Christoph E Hagemeyer 

Academic Editor

PLOS ONE